# Prognostic factors for outcomes of idiopathic sudden sensorineural hearing loss: protocol for the SeaSHeL national prospective cohort study

Rishi Mandavia [1,2,3] Gerjon Hannink,[4] Muhammad Nayeem Ahmed,[2,5] Yaami Premakumar,[6] Timothy Shun Man Chu,[3,7] Helen Blackshaw,[1,2] Tanjinah Ferdous,[1,2] Nishchay Mehta,[1,2] Joseph Manjaly,[1,2] Maha Khan,[8] Anne GM Schilder,[1,2] on behalf of the SeaSHeL Collaborative

For numbered affiliations see end of article.

**Correspondence to**
Mr Rishi Mandavia;
r.mandavia@ucl.ac.uk

## ABSTRACT

**Introduction** The mainstay of treatment for idiopathic sudden sensorineural hearing loss (SSNHL) includes oral steroids, intratympanic steroid injections or a combination of both. The National Institute for Health and Care Excellence, in their recent hearing loss guidelines, highlighted the paucity of evidence assessing the comparative effectiveness of these treatments; and the National Institute for Health Research (NIHR) Health Technology Assessment Programme has since released a commissioned call for a trial to identify the most effective route of administration of steroids as a first-line treatment for idiopathic SSNHL. For such trials to be run effectively, reliable information is needed on patients with SSNHL: where they present, numbers, demographics, treatment pathways, as well as outcomes. This study will collect these data in a nationwide cohort study of patients presenting with SSNHL across 97 National Health Service (NHS) trusts. The study will be delivered through ear, nose and throat (ENT) trainee networks, the NIHR Clinical Research Network (CRN) Audiology Champions and the NIHR CRN. Importantly, this study will also provide a dataset to develop a prognostic model to predict recovery for patients with idiopathic SSNHL. The study objectives are to: (1) map the patient pathway and identify the characteristics of adult patients presenting to NHS ENT and hearing services with SSNHL, (2) develop a prognostic model to predict recovery for patients with idiopathic SSNHL and (3) establish the impact of idiopathic SSNHL on patients' quality of life (QoL).
**Methods and analysis** Study design: national multicentre prospective cohort study across 97 NHS trusts.
Inclusion criteria: adult patients presenting to NHS ENT and hearing services with SSNHL.
Outcomes: change in auditory function; change in QoL score.
Analysis: multivariable prognostic model, using prespecified candidate predictors. Mean change in QoL scores will be calculated from initial presentation to follow-up.
**Ethics and dissemination** Health Research Authority and NHS Research Ethics Committee approved the study. Publication will be on behalf of study sites and collaborators.

## Strengths and limitations of this study

► Taking place across 97 National Health Service trusts, the SeaSHeL Study represents the first nationwide prospective cohort study of adults presenting to ear, nose and throat (ENT) and hearing services with sudden sensorineural hearing loss (SSNHL).

► It is uniquely delivered by National ENT Trainee Research Network, Student and Foundation Doctors in Otolaryngology, National Institute for Health Research (NIHR) Clinical Research Network (CRN) Audiology Champions and the NIHR CRN.

► We acknowledge that duration between onset of SSNHL and presentation to ENT services as well as follow-up times and treatment strategies will vary across centres and cases, which may impact measured outcomes; however identifying differences in the patient pathway will form an important study finding.

**Trial registration number** ClinicalTrials.gov Registry (NCT04108598).

## INTRODUCTION

Each year approximately 20 people per 100 000 experience a sudden hearing loss in one ear that is located in the inner ear.[1] This is called sudden sensorineural hearing loss (SSNHL), defined as a rapid loss of hearing occurring over 3 days, of 30 dB (decibel) or more, over at least three contiguous sound frequencies.[2] SSNHL is predominantly unilateral and the degree of hearing loss can range from mild to profound. In 71%–90% of cases, the cause is unknown despite investigation, and these cases are termed idiopathic SSNHL.[3 4]

Idiopathic SSNHL is a serious condition, adversely impacting people's lives[5] with

research showing associations with emotional distress, depression, difficulties at work and impaired social integration.[6 7] Its prognosis is poorly understood and may depend on age, comorbidities, degree of hearing loss, audiometric configuration, presence of vertigo, treatment received, time between onset of hearing loss and treatment, and other factors. It is estimated that 32%–65% of cases of idiopathic SSNHL recover spontaneously, although clinical experience suggests that this may be an overestimation, with further research required in this area.[2 8 9]

Current care pathways for patients suffering from SSNHL appear to vary considerably in terms of the type of service patients first present to, their subsequent referral, length of time between onset of symptoms, presentation and start of treatment, the treatment plan as well as follow-up. The mainstay of treatment options for idiopathic SSNHL include oral steroids, intratympanic steroid injections or a combination of both.[5] The National Institute for Health and Care Excellence, in their recent guidelines on hearing loss (NG98), highlighted the paucity of evidence assessing the comparative effectiveness of these treatments and issued a high priority research recommendation to identify the most effective route of administration of steroids as a first-line treatment for idiopathic SSNHL.[10] The National Institute for Health Research (NIHR) Health Technology Assessment Programme has since released a commissioned call for a trial to address this question.[11]

For such a trial to be run effectively, reliable information is needed on patients with SSNHL: where they present, numbers, demographics, treatment pathways, as well as outcomes. This study will collect these data in a nationwide prospective cohort study of adult patients presenting with SSNHL across 97 National Health Service (NHS) trusts. The study will be delivered through National ENT Trainee Research Network (INTEGRATE), Student and Foundation Doctors in Otolaryngology (SFO-UK), NIHR Clinical Research Network (CRN) Audiology Champions, and the NIHR CRN. Importantly, this study will not only provide key data to inform trial design and delivery, but also a unique dataset to develop a prognostic model to predict recovery for patients with idiopathic SSNHL.

## Objectives
1. To map the patient pathway and identify the characteristics of adult patients presenting to NHS ear, nose and throat (ENT) and hearing services with SSNHL.
2. To develop a prognostic model to predict recovery for patients with idiopathic SSNHL.
3. To establish the impact of idiopathic SSNHL on patients' quality of life (QoL).

## METHODS AND ANALYSIS
### Study design
National multicentre prospective observational cohort study. The study will be reported in accordance with

Strengthening the Reporting of Observational Studies in Epidemiology and 'Transparent Reporting of a multivariable prognostic model for Individual Prognosis or Diagnosis' reporting guidelines for observational studies.[12]

### Setting
A multicentre study taking place across 97 NHS trusts providing ENT and hearing services.

### Inclusion criteria
► Adult patients (man or woman) aged over 16 years of age presenting to NHS ENT and hearing services with a history of sudden hearing loss (within a 72-hour window) that is sensorineural in nature.
► There is a decrease in hearing of ≥30 dB affecting at least three consecutive frequencies (between the following specific frequencies: 250 Hz, 500 Hz, 1000 Hz, 2000 Hz, 4000 Hz and 8000 Hz), using a premorbid pure tone audiogram as a baseline reference. Or should a baseline pure tone audiogram not be available, the unaffected ear can be used a baseline reference. Or should both ears be affected by hearing loss, a normative reference (ISO7029)[13] can be used as the baseline reference.
► Willing and able to provide written informed consent.

### Exclusion criteria
► Patients with mixed or conductive hearing loss (CHL) in ear(s) with the sudden hearing loss. CHL will be defined as a 'true' air-bone gap of 15 dB or more in 3 or more contiguous frequencies, between the following specific frequencies: 500 Hz, 1000 Hz, 2000 Hz and 4000 Hz.

### Sample size and recruitment
The minimum sample size for developing a multivariable prognostic model to predict recovery for patients with idiopathic SSNHL was estimated using the method reported by Riley et al.[14] Assuming a prevalence (ie, full recovery of idiopathic SSNHL) of between 32% and 65%,[2 8 9] a Nagelkerke's R-squared of 0.5,[14] a 0.05 margin of error in estimation of intercept, and a shrinkage factor of ≥0.95, with our prespecified number of predictor parameters of 15, resulted in a minimum sample size of 510 patients. Assuming that 70% of patients with SSNHL are idiopathic cases, this gives a total study minimal sample size of 730 patients.[3 4]

Ninety-seven NHS trusts across the UK have been identified as study sites via INTEGRATE, SFO-UK, NIHR CRN Audiology Champions and the NIHR CRN. An audit carried out in preparation of this study at five NHS trusts, over 6 months, revealed an average presentation rate of one patient with idiopathic SSNHL per trust per month. It appears therefore that our recruitment target over the study duration (32 months) is feasible.

### Candidate predictors for prognostic model
Age; gender; social class; presence of vestibular symptoms; precipitating illness; pattern of hearing loss; severity of

hearing loss; time between onset of symptoms and treatment; treatment(s) received.

Pattern of hearing loss will be defined as: low frequency loss (250 Hz, 500 Hz and 1000 Hz), mid-frequency loss (500 Hz, 1000 Hz and 2000 Hz), high-frequency loss (2000 Hz, 4000 Hz and 8000 Hz) and all frequencies. Severity of hearing loss (on presentation) will be defined as follows: pure tone average (PTA) across six frequencies (250 Hz, 500 Hz, 1000 Hz, 2000 Hz, 4000 Hz and 8000 Hz), classified as: mild (25–40 dB loss), moderate (41–70 dB loss), severe (71–95 dB loss) and profound (>95 dB loss).

### Primary outcome

The change in auditory function in the affected ear from initial presentation to follow-up (at any one time between 6 and 16 weeks from onset of symptoms). Auditory function will be defined as the PTA of air conduction thresholds between the following specific frequencies: 250 Hz, 500 Hz, 1000 Hz, 2000 Hz, 4000 Hz and 8000 Hz. If multiple pure tone audiograms have been carried out between 6 and 16 weeks, the most recent pure tone audiogram will be used for the calculation of auditory function.

Change in auditory function classified as:

1. Full recovery
   – Final PTA in affected ear within 10 dB (≤10 dB) of PTA of baseline pure tone audiogram.
   – OR should a baseline pure tone audiogram not be available:
   – Final PTA in affected ear within 10 dB of PTA of unaffected ear.
   – OR should both ears be affected with hearing loss:
   – Final PTA in affected ear within 10 dB of a normative reference (ISO7029).
2. Partial to no recovery
   – Final PTA in affected ear ≥10 dB of PTA of baseline pure tone audiogram.

– OR should a baseline pure tone audiogram not be available:
– Final PTA in affected ear ≥10 dB of PTA of unaffected ear.
– OR should both ears be affected with hearing loss:
– Final PTA in affected ear ≥10 dB of PTA of a normative reference (ISO7029).[13]

### Secondary outcomes

1. Degree of change in auditory function
► Complete recovery
   – Final PTA in affected ear within 10 dB (≤10 dB) of PTA of baseline pure tone audiogram.
   – OR should a baseline pure tone audiogram not be available:
   – Final PTA in affected ear within 10 dB of PTA of unaffected ear (≤10 dB).
   – OR should both ears be affected with hearing loss:
   – Final PTA in affected ear within 10 dB of a normative reference (ISO7029).[13]
► Marked recovery
   – PTA improvement ≥30 dB (and final PTA in affected ear ≥10 dB of PTA of baseline pure tone audiogram).
   – OR should a baseline pure tone audiogram not be available:
   – PTA improvement ≥30 dB (and final PTA in affected ear ≥10 dB of PTA of unaffected ear).
   – OR should both ears be affected with hearing loss:
   – PTA improvement ≥30 dB (and final PTA in affected ear ≥10 dB of a normative reference (ISO7029)).[13]
► Slight recovery
   – PTA improvement ≥10 dB and <30 dB (and final PTA in affected ear ≥10 dB of PTA of baseline pure tone audiogram).

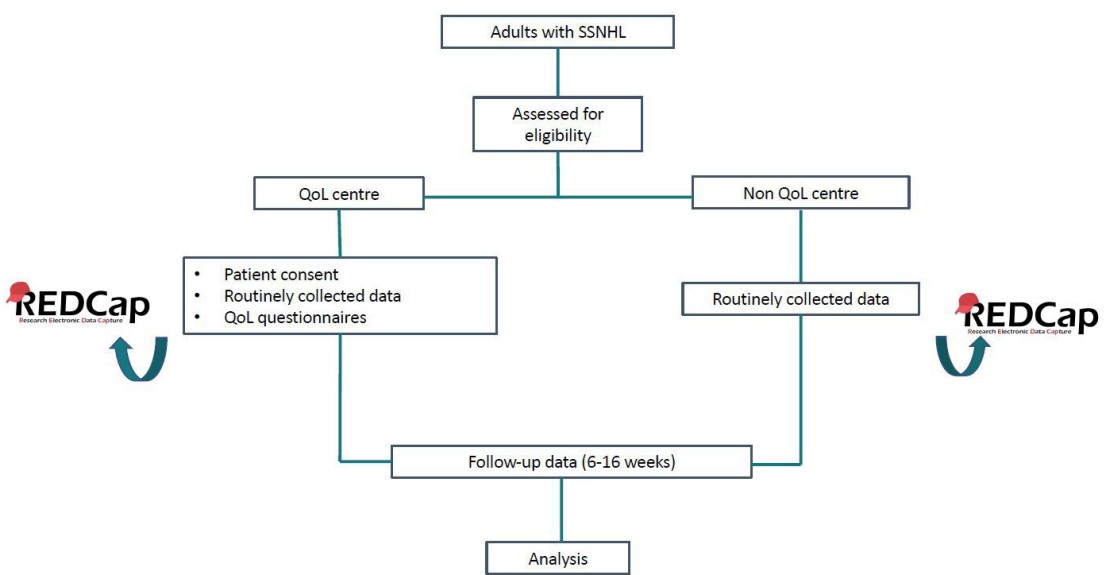

**Figure 1** Summary of study procedures. A brief outline of study procedures from patient recruitment to data analysis. QoL, quality of life; SSNHL, sudden sensorineural hearing loss.

– OR should a baseline pure tone audiogram not be available:
– PTA improvement ≥10 dB and <30 dB (and final PTA in affected ear ≥10 dB of PTA of unaffected ear).
– OR should both ears be affected with hearing loss:
– PTA improvement ≥10 dB and <30 dB (and final PTA in affected ear ≥10 dB of a normative reference (ISO7029)).[13]

► No improvement
– PTA improvement <10 dB (and final PTA in affected ear ≥10 dB of PTA of baseline pure tone audiogram).
– OR should a baseline pure tone audiogram not be available:
– PTA improvement <10 dB (and final PTA in affected ear ≥10 dB of PTA of unaffected ear).
– OR should both ears be affected with hearing loss:
– PTA improvement of <10 dB (and final PTA in affected ear ≥10 dB of a normative reference (ISO7029)).[13]

2. Change in QoL score from initial presentation to follow-up at any one time between 6 and 16 weeks following treatment. QoL will be measured with the Hearing Handicap Inventory for Adults (HHIA) (for patients under 65 years of age) or Hearing Handicap Inventory for Elderly (HHIE) (for patients over 65 years of age) and the Health Utility Index Mark 3 (HUI3). QoL data will be collected from a subset of 20 trusts.

Out-of-limit thresholds obtained on the pure tone audiogram will be classified as a 96 dB loss at that specific frequency.

## Statistical analysis

We will develop a prognostic model to predict recovery for patients with idiopathic SSNHL and assess the impact of idiopathic SSNHL on patients' QoL. A multivariable prognostic model will be developed using the candidate predictors specified above (n=15). Missing outcome data at the end of the study will be imputed using multiple imputation by a chained equations procedure.[15] Internal validation will be performed to quantify optimism in the predictive performance (calibration and discrimination) of the developed model using bootstrapping techniques. Bootstrapping techniques provide information on the performance of the model in comparable datasets and generate a shrinkage factor to adjust the regression coefficients.[16 17]

The mean change in HHIA, HHIE and HUI3 scores across patients with idiopathic SSNHL will be calculated from initial presentation to follow-up (any one time between 6 and 16 weeks). We will use the non-parametric Wilcoxon and McNemar–Bowker tests with a significance level of 5%. Statistical analysis will be carried out using R V.3.6.2 (R Foundation for Statistical Computing, Vienna, Austria).

## Study procedures

A summary of study procedures is outlined in figure 1. Box 1 provides a summary of all data to be collected. Adult patients with SSNHL presenting to any of the 97 NHS trusts will be assessed for eligibility. In 20 of 97 NHS trusts, all data identified in box 1 will be collected. In the remaining 77 trusts, all data except for QoL data will be collected. In centres where no QoL data are collected, there will be no deviation from the usual standard of care for any patient, and routinely collected patient data will be anonymised; therefore, consent is not required. For centres where QoL data are collected, patients will be provided with a study information sheet and requested to give written consent to participate in the study.

Patient follow-up data will be collected at 6–16 weeks from onset of symptoms. All data collected will be uploaded onto REDCap, a secure online data collection platform.[18 19] All data will be transferred and stored

---

**Box 1  Data to be collected**

► General information.
► Patient demographics and socioeconomic status.
► Medical history and social history.
► Patient's first presentation to ear, nose and throat (ENT).
► Hospital and service patient presenting to.
► Date of presentation.
► Referral pathway and treatments received up until first presentation to ENT.
► Clinical history of sudden sensorineural hearing loss (SSNHL).
► Examination findings.
► Audiometry results.
► Imaging requests/results.
► Laboratory test requests/results (data on any laboratory test requested for a patient with SSNHL to be collected).
► Vestibular function tests/results (data on any vestibular function test requested for a patient with SSNHL to be collected).
► Hearing Handicap Inventory for Adults (HHIA) Questionnaire score OR Hearing Handicap Inventory for Elderly (HHIE) Questionnaire score.*
► Health Utility Index Mark 3 (HUI3) Questionnaire score.*
► Treatment plan.
► Cause of SSNHL identified.
► Follow-up details.
► Follow-up (6–16 weeks after onset of symptoms).
► Date of presentation.
► Examination findings.
► Audiometry results.
► Imaging requests/results.
► Laboratory test results (data on any laboratory test requested for a patient with SSNHL to be collected).
► Vestibular function results (data on any vestibular function test requested for a patient with SSNHL to be collected).
► HHIA Questionnaire score OR HHIE Questionnaire score.*
► HUI3 Questionnaire score.*
► Treatments provided since first visit to ENT.
► Cause of SSNHL identified.
► Adverse events.
*To be collected at quality of life sites only (20 of 97 sites).

---

in accordance with data governance regulations (Data Protection Act 2018).

## Patient and public involvement

Three patients suffering from SSNHL were involved in the design of this study from its inception. They were recruited from a database of patients interested in taking part in Patient and Public Involvement activities. They reviewed and gave input on study objectives and design, including reviewing participant information sheets and consent forms to ensure clarity. This group will also be involved in monitoring study progress and in its dissemination including coauthoring a lay summary report.

## ETHICS AND DISSEMINATION
### Ethics approval

Health Research Authority and NHS Research Ethics Committee has approved the study. IRAS project ID: 258494; REC reference: 19/NW/0556.

Risks, Burdens and Benefits

There are no anticipated risks or direct benefits to participants.

### Dissemination

Results will be presented at local and inter(national) meetings and published in the scientific and grey literature for the attention of professional and scientific audiences. Publication and presentation of the final results will be on behalf of all study sites and collaborators. A lay summary report will be published for patients and members of the public.

### Study start and end date

Start date: 7 October 2019.

Planned end date: 1 May 2022.

**Author affiliations**

[1]University College London Hospitals Biomedical Research Centre, National Institute for Health Research, London, UK
[2]evidENT, University College London Ear Institute, London, UK
[3]SFO UK Students and Foundation Doctors in Otolaryngology, London, UK
[4]Department of Operating Rooms, Radboudumc, Nijmegen, The Netherlands
[5]University of Leeds School of Medicine, Leeds, UK
[6]Chelsea and Westminster Hospital, London, UK
[7]Newcastle University School of Clinical Medical Sciences, Newcastle upon Tyne, UK
[8]Health Education North West, Manchester, UK

**Collaborators** The SeaSHeL.

**Contributors** The SeaSHeL Academic team were involved in writing this protocol. RM, YP, NM, TF, HB, MK, JM, GH, TSMC, MNA and AS made substantial contributions to the conception and design of the work. RM, YP, NM, TF, HB, MK, JM, GH, TSMC, MNA and AS were involved in drafting the article and revising it critically. RM, YP, NM, TF, HB, MK, JM, GH, TSMC, MNA and AS approved the final version to be published. RM, YP, NM, TF, HB, MK, JM, GH, TSMC, MNA and AS agree to be accountable for all aspects of the work in ensuring that questions related to the accuracy or integrity of any part of the work are appropriately investigated and resolved.

**Funding** RM, AS, HB, NM, JM and TF are supported by the NIHR UCLH BRC (Award number 254). Sensorion provided a small grant towards administrative support (Grant number S6836).

**Disclaimer** Sensorion has not been involved in the design of this study, nor will have access to the study data, be involved in data analysis or write-up of results.

**Competing interests** Sensorion is an inner ear disease company and sponsor of the NIHR Clinical Research Network Portfolio Study, Audible-S (CPMS 39560, IRAS 248645).

**Patient and public involvement** Patients and/or the public were involved in the design, or conduct, or reporting, or dissemination plans of this research. Refer to the Methods section for further details.

**Patient consent for publication** Not required.

**Provenance and peer review** Not commissioned; externally peer reviewed.

**ORCID iD**
Rishi Mandavia http://orcid.org/0000-0002-5839-2735

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
