## [Reviewer comments · BMJ Open]

ARTICLE DETAILS

TITLE (PROVISIONAL)	Prognostic factors for outcomes of idiopathic Sudden Sensorineural Hearing Loss: protocol for the SeaSheL national prospective cohort study
AUTHORS	Mandavia, Rishi; Hannink, Gerjon; Ahmed, Muhammad; Premakumar, Yaami; Chu, Timothy; Blackshaw, Helen; Ferdous, Tanjinah; Mehta, Nishchay; Manjaly, Joseph; Khan, Maha; Schilder, Anne

VERSION 1 – REVIEW

REVIEWER	Torsten Rahne University Hospital Halle (Saale), Dept. of Otorhinolaryngology, University Medicine Halle (Saale), Germany
REVIEW RETURNED	27-Mar-2020

GENERAL COMMENTS	Study protocol: Prognostic factors for outcomes of idiopathic Sudden Sensorineural Hearing Loss: protocol for the SeaSheL national prospective cohort study Review The study protocol describes an ambitious enterprise. Data recorded by 97 national centers will be collected and analyzed. The study is worthwhile, important and of high scientific relevance. The protocol is well written, concise and well designed. However, some drawbacks should be fixed before starting the trial. Overall, the discrimination between SSNHL and idiopathic SSNHL (iSSNHL) has been mixed up throughout the protocol. P7L28 says that iSSNHL is a subgroup of SSNHL (71-90%). Thus, the protocol should be define very clearly, which subgroup will be included in the study. Do you focus on iSSNHL only? Than your subgroup would depend on the type of measure, e.g., from doing a MRI to diagnose a vestibular or schwannoma or not. Please clarify the objectives, inclusion criteria respectively and modify the case number estimation appropriately. Objectives: Measuring QoL immediately after a SSNHL would be cause a high variance of the result, since the impact of the SSNHL on te QoL might be judged very differently by the subjects after only a few days of experience with that situation. It is unclear how the SSNHL will be measured as inclusion criterion and used as outcome parameter: The protocol allows two affected ears, but the change in auditory function is measured using the contralateral ear as reference. I
---

	urgently advise to use a graded criterion: 1) use the audiogram before the SSNHL. Since it will not be available in many cases, 2) use the audiogram of the contralateral ear. If the contralateral ear shows a CHL or (S)SNHL as well, 3) use a normative reference (ISO7029). P9L48: What does 30 dB HL exactly means? Do you mean the absolute threshold (dB HL) or the difference (i.e., the SSNHL) compared to a reference threshold? P9L50, P10L6 and P11L15 please define averaged frequencies exactly (“between“ or “at”) P10L3: It is unclear, how CHL in the contralateral ear is handled. Exclusion? P10L6: please remove “HL”, since the ABG is a difference measure. The sample size calculation reveals a plausible number of participants. However, an estimation if and how this number could be achieved is missing. Furthermore, I assume that the outcome parameter will be stratified by the covariates. It is unclear how exactly this stratification will be performed and how this has influenced the calculation of case numbers. P10L18: Please check citation of reference [12] (it is not “Riley et al.”) Pattern of hearing loss: Please clarify how mixed patterns (e.g., low and high) will be handled. How will hearing thresholds “out of limit” be handled? How is the PTA measured in those cases? Statistics: Why do you intend to use two software solutions for the statistics (R and SPSS)? May the guidelines of BMJ open requires it differently, but to me the dots should always be put after the literature reference, e.g.: “example text [5].”
--	--

REVIEWER	Jae Ho Chung Hanyang University Republic of Korea
REVIEW RETURNED	19-Apr-2020

GENERAL COMMENTS	The authors have to consider the duration between the onset of hearing loss and treatment start. Please include the possible prognostic factors for ISSNHL. In addition, this is protocol did not include laboratory data or vestibular function tests. I think the authors have to use objective parameters rather than history of vertigo. Management strategies must be included in the protocol. In this multicenter study, heterogeneity of management methods might disturb the study results.
--

VERSION 1 – AUTHOR RESPONSE

REVIEWER 1

COMMENT:

The study protocol describes an ambitious enterprise. Data recorded by 97 national centers will be collected and analyzed.

The study is worthwhile, important and of high scientific relevance. The protocol is well written, concise and well designed.

However, some drawbacks should be fixed before starting the trial.

Overall, the discrimination between SSNHL and idiopathic SSNHL (iSSNHL) has been mixed up throughout the protocol. P7L28 says that iSSNHL is a subgroup of SSNHL (71-90%). Thus, the protocol should be define very clearly, which subgroup will be included in the study. Do you focus on iSSNHL only? Than your subgroup would depend on the type of measure, e.g., from doing a MRI to diagnose a vestibular or schwannoma or not. Please clarify the objectives, inclusion criteria respectively and modify the case number estimation appropriately.

RESPONSE:

Thank you for your valuable comments and apologies for the lack in clarity.

In the manuscript, we have now clarified that our objectives are:

1. To map the patient pathway and identify the characteristics of adult patients presenting to NHS ENT and hearing services with SSNHL. (This will include all patients with SSNHL, not only idiopathic cases)
2. To develop a prognostic model to predict recovery for patients with idiopathic SSNHL.
3. To establish the impact of idiopathic SSNHL on patients' quality of life (QoL).

We have clarified in the inclusion criteria that we are including all adults that present with SSNHL (not only idiopathic cases). From the data collected, we will be able to determine which patients were diagnosed with idiopathic SSNHL (see Table 1)

We have also clarified in the analysis that we will develop a prognostic model to predict recovery for patients with idiopathic SSNHL; and assess the impact of idiopathic SSNHL on patients' quality of life (QoL).

We have now modified the case number estimation accordingly to 730 patients. Please see section "Sample size and recruitment".

COMMENT

Objectives: Measuring QoL immediately after a SSNHL would be cause a high variance of the result, since the impact of the SSNHL on the QoL might be judged very differently by the subjects after only a few days of experience with that situation.

RESPONSE:

Thank you for this point which we certainly agree with. From a practical perspective, it is uncertain when patients will first present to ENT services following experiencing SSNHL, and as a result, the number of days post onset when QoL will be measured will vary, which may result in a variance of outcome. We have included this limitation in the limitations section.

COMMENT:

It is unclear how the SSNHL will be measured as inclusion criterion and used as outcome parameter: The protocol allows two affected ears, but the change in auditory function is measured using the contralateral ear as reference. I urgently advise to use a graded criterion: 1) use the audiogram before the SSNHL. Since it will not be available in many cases, 2) use the audiogram of the contralateral ear. If the contralateral ear shows a CHL or (S)SNHL as well, 3) use a normative reference (ISO7029).

RESPONSE:

Thank you for raising this valuable point. As suggested we have now adopted a graded criterion for our primary and secondary outcomes.

COMMENT:

P9L48: What does 30 dB HL exactly means? Do you mean the absolute threshold (dB HL) or the difference (i.e., the SSNHL) compared to a reference threshold?

RESPONSE:

This refers to absolute threshold. We have now clarified this in the manuscript.

COMMENT:

P9L50, P10L6 and P11L15 please define averaged frequencies exactly ("between" or "at")

RESPONSE:

Thank you, this has now been more clearly defined.

COMMENT:

P10L3: It is unclear, how CHL in the contralateral ear is handled. Exclusion?

RESPONSE:

We have now clarified that patients with conductive hearing loss (CHL) in the presenting ear(s) will be excluded. Patients with CHL in the contralateral ear can be included.

COMMENT:

P10L6: please remove "HL", since the ABG is a difference measure.

RESPONSE:

Thank you, this has been removed.

COMMENT:

The sample size calculation reveals a plausible number of participants. However, an estimation if and how this number could be achieved is missing.

RESPONSE:

We have now discussed in the 'Sample size and recruitment' section that an audit was carried out in preparation of this study at 5 NHS trusts, over 6 months which revealed an average presentation rate of 1 patient with idiopathic SSNHL per trust per month. It appears therefore that our recruitment target over the study duration (32 months) is feasible.

COMMENT

Furthermore, I assume that the outcome parameter will be stratified by the covariates. It is unclear how exactly this stratification will be performed and how this has influenced the calculation of case numbers.

RESPONSE:

Thanks for this comment and apologies for the lack in clarity. We have now clarified in the 'Sample size and recruitment' section that the objective is to develop a prognostic model to predict recovery for patients with idiopathic SSNHL, not to stratify the outcome by covariates. Therefore, sample size calculation focusses on prognostic model development. Recent work by Riley et al. (reference 13) provides guidance on how to calculate the required sample size for prediction model development. We followed these guidelines in our sample size calculations as specified in the 'Sample size and recruitment' section.

COMMENT:

P10L18: Please check citation of reference [12] (it is not "Riley et al.")

RESPONSE:

Thanks for bringing this to our attention. We have corrected this.

COMMENT:

Pattern of hearing loss: Please clarify how mixed patterns (e.g., low and high) will be handled.

RESPONSE:

This has now been clarified in the section "Candidate predictors for prognostic model": Pattern of hearing loss will be defined as: low frequency loss (250, 500, 1000 Hz), mid frequency loss (500, 1000, 2000 Hz), high frequency loss (2000, 4000 Hz and 8000 Hz) and all frequencies.

Patterns of hearing loss will be used as candidate predictors for the prognostic model. This has been clarified in the section 'Candidate predictors of prognostic model'

COMMENT:

How will hearing thresholds "out of limit" be handled? How is the PTA measured in those cases?

RESPONSE:

Out of limit thresholds obtained on the pure tone audiogram will be classified as a Profound, 96 dB loss at that specific frequency. This has been specified on page 14

COMMENT

Statistics: Why do you intend to use two software solutions for the statistics (R and SPSS)?

RESPONSE:

Thank you for raising this. We will only use R. We have edited the section "Statistical analysis" accordingly.

Reviewer: 2

Reviewer Name: Jae Ho Chung

Institution and Country:

Hanyang University

Republic of Korea

Please state any competing interests or state 'None declared': none

COMMENT:

The authors have to consider the duration between the onset of hearing loss and treatment start.

RESPONSE:

Thanks for your comment. We acknowledge that duration between onset of hearing loss and treatment start will differ across centres and cases which may impact measured outcomes. We have included this in the limitation section, however note that identifying such differences in the patient pathway will be an important study finding.

COMMENT:

Please include the possible prognostic factors for ISSNHL.

RESPONSE:

Thanks for your comment. These have been included in the section "candidate predictors for prognostic model".

COMMENT:

In addition this protocol did not include laboratory data or vestibular function tests. I think the authors have to use objective parameters rather than history of vertigo.

RESPONSE:

We have clarified in Table 1 a summary of all data that will be collected. This includes laboratory tests and vestibular function tests.

COMMENT:

Management strategies must be included in the protocol. In this multicenter study, heterogeneity of management methods might disturb the study results.

RESPONSE:

Thanks for this comment. One of the main objectives of this study is to map the patient pathway of adult patients presenting with SSNHL, which will include capturing variance in management strategies. The management strategy for each patient will be recorded as specified in Table 1. We acknowledge that management methods may be heterogenous across centres, which may impact measured outcomes. We have included this in the limitation section, however highlight that identifying such differences in the patient pathway will be an important study finding.

COMMENT:

Corresponding author email address in ScholarOne system is different from the main document. Kindly amend accordingly.

RESPONSE:

This has been amended

COMMENT:

- The spelling of the name of co-author's 'Yaami Premakumar, Nishchay Mehta, and M. M. Khan' in ScholarOne system are different from the main document. Kindly amend accordingly.

RESPONSE:

This has been amended

VERSION 2 – REVIEW

REVIEWER	Torsten Rahne University Hospital Halle (Saale), Germany
REVIEW RETURNED	23-Jul-2020

GENERAL COMMENTS	Definition of inclusion criteria ("Diagnosed with a hearing loss in one or both ears of 30 dB HL (absolute threshold)") is still unclear. If you keep this, you should set an age limit (e.g., 30 years old) to prevent inclusion of patients with age-related hearing loss. Or, alternatively, change the inclusion criterion to a relative measure of hearing loss, e.g., 30 dB referenced to a baseline. Please specify laboratory and vestibular tests.
--

REVIEWER	Jae Ho Chung Hanyang University Korea
REVIEW RETURNED	27-Jul-2020

GENERAL COMMENTS	Authors successfully revised the manuscript according to the reviewer's suggestions.
--

VERSION 2 – AUTHOR RESPONSE

REVIEWER 1 COMMENTS:

COMMENT:

Definition of inclusion criteria ("Diagnosed with a hearing loss in one or both ears of 30 dB HL (absolute threshold)") is still unclear. If you keep this, you should set an age limit (e.g., 30 years old) to prevent inclusion of patients with age-related hearing loss.

Or, alternatively, change the inclusion criterion to a relative measure of hearing loss, e.g., 30 dB referenced to a baseline.

RESPONSE:

Thanks for your comment. We agree that this may result in ambiguity.

We have now specified the following:

"Inclusion criteria:

- Adult patients (male or female) aged over 16 years of age presenting to NHS ENT and hearing services with a history of sudden hearing loss (within a 72-hour window) that is sensorineural in nature.

AND

There is a decrease in hearing of ≥ 30 dB affecting at least 3 consecutive frequencies (between the following specific frequencies: 250, 500, 1000, 2000, 4000 and 8000 Hz), using a pre-morbid pure tone audiogram as a baseline reference. OR should a baseline pure tone audiogram not be available, the unaffected ear can be used as a baseline reference. Or should both ears be affected by hearing loss, a normative reference (ISO7029)[13] can be used as the baseline reference.

AND

- Willing and able to provide written informed consent.

COMMENT:

Please specify laboratory and vestibular tests.

RESPONSE:

We have now clarified that any laboratory or vestibular test requested for a patient with SSNHL be recorded, in keeping with our objective of identifying differences in the patient pathway.

REVIEWER 2 COMMENT:

COMMENT:

Authors successfully revised the manuscript according to the reviewer's suggestions.

RESPONSE:

Thanks again for your valuable comments during the review process.